# Building Height Extraction from GF-7 Satellite Images Based on Roof Contour Constrained Stereo Matching

**Chenni Zhang** [1], **Yunfan Cui** [2], **Zeyao Zhu** [2], **San Jiang** [3] and **Wanshou Jiang** [1,*]

1 State Key Laboratory of Information Engineering in Surveying Mapping and Remote Sensing, Wuhan University, Wuhan 430079, China; 2019206190070@whu.edu.cn
2 School of Remote Sensing & Information Engineering, Wuhan University, Wuhan 430079, China; yunfancui@whu.edu.cn (Y.C.); rszyzhu@whu.edu.cn (Z.Z.)
3 School of Computer Science, China University of Geosciences, Wuhan 430074, China; jiangsan@cug.edu.cn
* Correspondence: jws@whu.edu.cn; Tel.: +86-27-6877-8424

**Abstract:** Building height is one of the basic geographic information for planning and analysis in urban construction. It is still very challenging to estimate the accurate height of complex buildings from satellite images, especially for buildings with podium. This paper proposes a solution for building height estimation from GF-7 satellite images by using a roof contour constrained stereo matching algorithm and DSM (Digital Surface Model) based bottom elevation estimation. First, an object-oriented roof matching algorithm is proposed based on building contour to extract accurate building roof elevation from GF-7 stereo image, and DSM generated from the GF-7 stereo images is then used to obtain building bottom elevation. Second, roof contour constrained stereo matching is conducted between backward and forward image blocks, in which the difference of standard deviation maps is used for the similarity measure. To deal with the multi-height problem of podium buildings, the gray difference image is adopted to segment podium buildings, and re-matching is conducted to find out their actual heights. Third, the building height is obtained through the elevation difference between the building top and bottom, in which the evaluation of the building bottom is calculated according to the elevation histogram statistics of the building buffer in DSM. Finally, two GF-7 stereo satellite images, collected in Yingde, Guangzhou, and Xi'an, Shanxi, are used for performance evaluation. Besides, the aerial LiDAR point cloud is used for absolute accuracy evaluation. The results demonstrate that compared with other methods, our solution obviously improves the accuracy of height estimation of high-rise buildings. The MAE (Mean Absolute Error) of the estimated building heights in Yingde is 2.31 m, and the MAE of the estimated elevation of building top and bottom is approximately 1.57 m and 1.91 m, respectively. Then the RMSE (Root Mean Square Error) of building top and bottom is 2.01 m and 2.57 m. As for the Xi'an dataset with 7 buildings with podium out of 40 buildings, the MAE of the estimated building height is 1.69 m and the RMSE is 2.34 m. The proposed method can be an effective solution for building height extraction from GF-7 satellite images.

**Keywords:** GF-7 satellite; building height estimation; contour constrained matching; podium segmentation; urban 3D reconstruction

## 1. Introduction

With the acceleration of China's urbanization process, the concept of "urban fine management" has been proposed [1]. Urban fine management is designed to solve the problems of blind expansion [2,3], population assessment [4–6], urban climate [7,8], destruction of natural environment and cultural heritage [9,10], urban 3D reconstruction [11–14], etc. As the fundamental element of urban cities, buildings are one of the most important research targets, and their height information plays a critical role in urban exemplary management. Thus, effective solutions are necessarily required in practice.

In contrast to shadow-based methods [15–22], other research aims to use dense matching technology to obtain DSM (Digital Surface Model) of building areas [23–26], in which building height can be calculated from the roof and bottom elevation of buildings [27,28]. The methods based on DSM have become the most commonly used solution, and the building height accuracy depends on the accuracy of the roof and bottom elevation. In general, there are two kinds of DSM extraction methods. One is the traditional dense matching methods, such as SGM (semi-global matching) [29]. The other is the deep learning-based methods [30–34]. The SGM algorithm combines the advantages of the local algorithm and the global algorithm, and avoids the disadvantages of them [29,35]. It has been adopted by most commercial satellite image processing software. For example, Tao et al. [36] used PCI and INPHO to explore DSM reconstruction of GF-7 images, and the accuracy of flat area can reach 0.655 m. However, the experimental data that was selected from Xinjiang with only a few low buildings, which cannot conclude for urban areas with dense high-rise buildings. Wang et al. [28] adopted SGM to generate point clouds from GF-7 stereo images for building height extraction, in which building footprints are extracted on the backward image by using a proposed multi-stage attention U-Net. The root mean square error (RMSE) between the extracted building height and the reference building height is 5.41 m, and the mean absolute error (MAE) is 3.39 m. The results demonstrate the promising potential of GF-7 RS images in stereo applications. However, they did not discuss the big forward angle of the GF-7 satellite camera, which leads to the loss of high-rise buildings when applying the pyramid SGM methods to estimate DSM.

On the other side, deep learning-based methods have been designed to extract elevation or height information from aerial or satellite images [37–40]. Karatsiolis S et al. [39] proposed a task-centered deep learning (DL) model, which combines the structural characteristics of the U-Net and residual network, and learns the mapping from single aerial image to standardized digital ground model. The model was trained on aerial images whose corresponding DSM and Digital Terrain Models (DTM) were available, and they were then used to infer the nDSM of images with no elevation information. To promote the deep learning based elevation and height information extraction from satellite images, IEEE released the worldview satellite images and corresponding DSM, DTM, and nDSM data [37]. Cao et al. [38] used ZY-3 data combined with A-map (a map service provider of China) building height data and proposed a multi-view, multi-spectral, and multi-objective neural network (called M3Net) to extract large-scale building footprints and heights on the backward image. He verified the applicability of the proposed method in various cities. The RMSE in the test site of Shenzhen is 6.43 m by M3Net. Although deep learning-based methods have shown exciting performance, we still can not use deep learning-based methods in the operational system, due to the requirement of a large number of sample data with ground truth for different satellites, different locations and different views. At present, there are few sample data databases for satellite image stereo matching. Due to the different design of different satellites in image spatial resolutions and stereo intersection angles, the sample database for one satellite cannot be used directly for other satellites [36].

Considering the above-mentioned issues, this paper proposes an object-oriented building roof matching method to improve the accuracy of building roof elevation of GF-7 images. Different from Wang et al. [28], who focused on the extraction of building footprints and directly used SGM for extracting building height, we focus on the extraction of roof elevation, in which DSM provides only bottom elevation information in the proposed method. In this paper, we do not discuss the extraction of building contour, but use the self-marked contours as input. In general, DSM and building contour are our input.

Our method consists of three steps: (1) The first step is image pre-processing, including the calculation of the forward image search range and feature map preparation. The image search range is determined by the minimum and maximum possible elevation of building roofs, which can be estimated with a DSM and the maximum possible height of the highest building. With the search range, the contour of the building on the backward image is projected to the forward image to obtain epipolar image blocks. (2) The second step is roof

contour constrained stereo matching and podium segmentation, in which the backward image block is matched against the forward image with the similarity metre of feature map difference. Since the podium contains multiple elevation planes, the podium is segmented with image gray difference and rematched to obtain the elevation of multiple planes. (3) The third step is bottom elevation estimation and building height calculation. The bottom elevation is calculated according to the elevation histogram statistics of building buffer by using DSM. The top elevation is calculated using a space intersection algorithm. The building height is then obtained through the difference between the top and bottom elevations.

This paper is organized as follows. Our methodology is presented in Section 2. The experiment data and results are reported in Section 3. Finally, conclusions are drawn in Section 4.

## 2. Methodology

This paper proposes a roof contour constrained stereo matching method to extract the accurate height of high-rise buildings from GF-7 satellite images. The overall workflow of the proposed solution is illustrated in Figure 1. The input of our method consists of GF-7 backward and forward images, building roof contours on the backward image, and the corresponding DSM data. The method includes three major steps, as described as follows:

(1) Epipolar image block and feature map generation from stereo images. The first step is to prepare the epipolar image block for roof contour matching, in which the parallax search range is determined by using the DSM data and a supposed maximum range of building heights. After image resampling, the epipolar image blocks are transformed into feature map blocks by computing regional standard deviation to extract the structural characteristics.

(2) Roof contour constrained stereo matching considering podium structure. The backward image block is first matched against the forward image block with the similarity measure of image feature difference. To deal with the multi-heights problem arising from podium buildings, the gray difference image is used to detect the mismatched contours, which can include podium. The image is then rematched iteratively until no podium structure can be detected. With the matched parallax, the roof elevation is calculated through the space intersection using the RPCs of satellite stereo images and the 2D roof contours on the backward image is transformed into 3D roof contours in the geographic coordinate system.

(3) Bottom elevation estimation and building height calculation. Bottom elevation is estimated using histogram analysis of the DSM in the buffer of a building contour with a buffer size of 20 m. Building height is then calculated by the subtraction between bottom elevation and roof elevation.

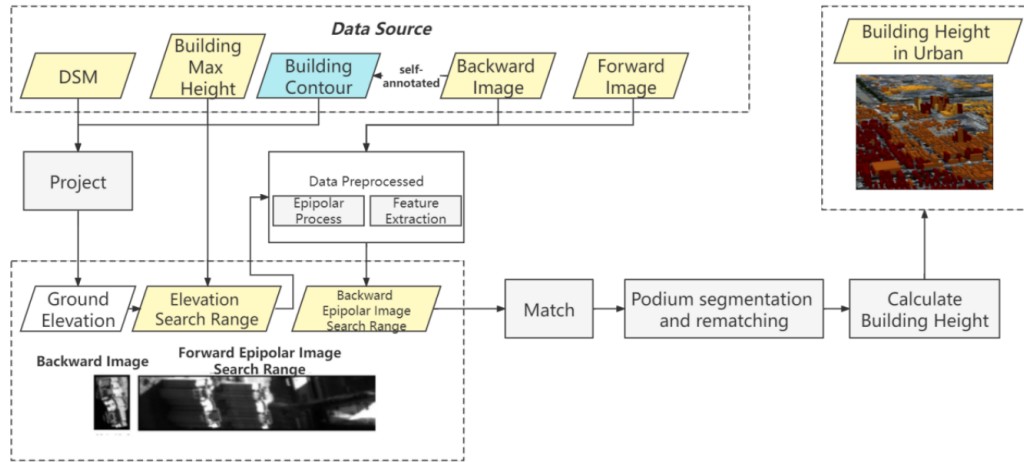

**Figure 1.** The overall workflow of the building height extraction solution.

### 2.1. Epipolar Image Block and Feature Map Generation for Stereo Images

This step aims to prepare epipolar image blocks for roof contour constrained stereo matching. The source block range on the backward image is determined by the building contour itself. The destination block range on the forward image depends on the building roof elevation, which needs to be determined by image matching. So we use DSM for estimating the bottom elevation and plus the maximum possible height for roof elevation. Then the roof contour is transformed from the backward image to the forward image for epipolar resampling. At last, convolution processing is used to extract the standard deviation feature of the epipolar image blocks for decreasing the impact of bright change between the forward image and the backward image.

#### 2.1.1. Calculation of Parallex Search Range

The principle of parallex search range is illustrated in Figure 2. The input building roof contour is on the 2D backward image, and the DSM is in the 3D object space. An iteration algorithm is used to obtain building roof elevation from the DSM data, in which it starts from an initial elevation $Z_0$, such as the mean elevation of the study area. Then, the algorithm executes iteratively as follows:

(1) With the initial evaluation $Z_0$, we can get point $P_0$ in the 3D object space by intersecting imaging ray through the roof center with the horizontal plane created by the elevation of $Z_0$.

(2) After obtaining point $P_0$, we use the horizontal coordinates X and Y of point $P_0$ to interpolate a new elevation value of $Z_1$ from the DSM data.

(3) If the elevation difference between $Z_1$ and $Z_0$ is greater than a given threshold, then update $Z_0$ with $Z_1$.

(4) Steps 1 to 3 are iteratively executed until reaching the terminal condition.

By using the above-mentioned algorithm, the minimum elevation can be obtained, which is termed as $Z_{min}$. Considering the maximum building height $h_{max}$ in the test area, the elevation search range can be calculated as $Z_{min} + h_{max}$. With the elevation search range $(Z_{min}, Z_{max})$, the parallex search range (begin, end) on forward image can be calculated by projecting the points of roof contours on the backward image to the forward image, whose envelope consists of the entire parallex search range on the forward image.

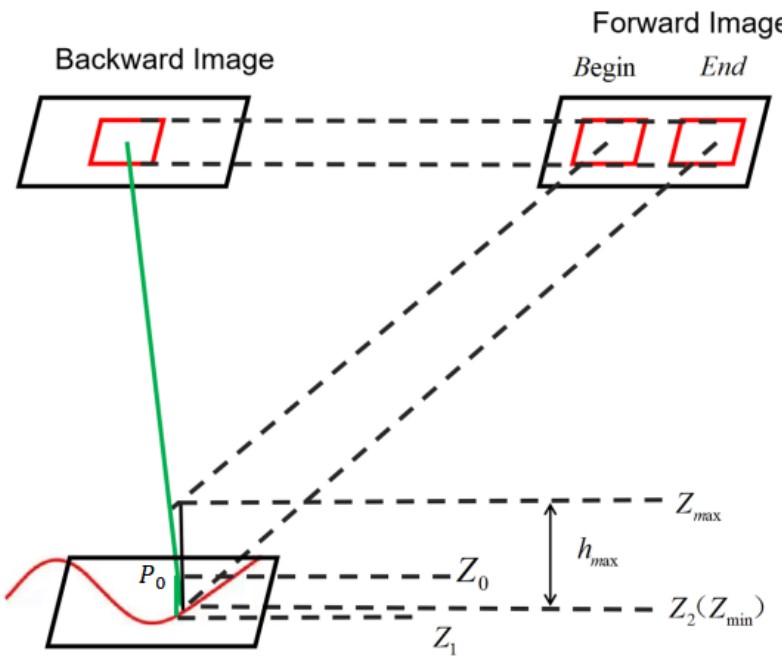

**Figure 2.** The principle of parallex search range calculation.

### 2.1.2. Epipolar Image Block and Feature Map Generation

By using the determined parallex search range, the GF-7 image is resampled by the epipolar transform to generate epipolar image blocks. After resampling the forward and backward images, their y parallax is eliminated. The two-dimensional image matching is simplified as one-dimensional matching in order to improve the matching efficiency [23,41].

The brightness change between the forward image and the backward image can greatly affect the image matching. To decrease the impact of brightness change, we conduct the image matching on the standard deviation maps instead of the original epipolar image block themselves. As shown in Figure 3, we use a convolution of $5 \times 5$ window to generate standard deviation maps. The larger the standard deviation value, the richer the image pixel information. The standard deviation is calculated with Equation (1)

$$S_{(i,j)} = \sqrt{\frac{\sum_{a=-n}^{n} \sum_{b=-n}^{n} (x_{(i+a,j+b)} - \bar{x})^2}{(2n+1)^2}} \tag{1}$$

where $S_{(i,j)}$ is the standard deviation value at point $(i,j)$; $n$ is the radius of the window; $\bar{x}$ is the mean value of the window; $x_{(i+a,j+b)}$ is the pixel in the window.

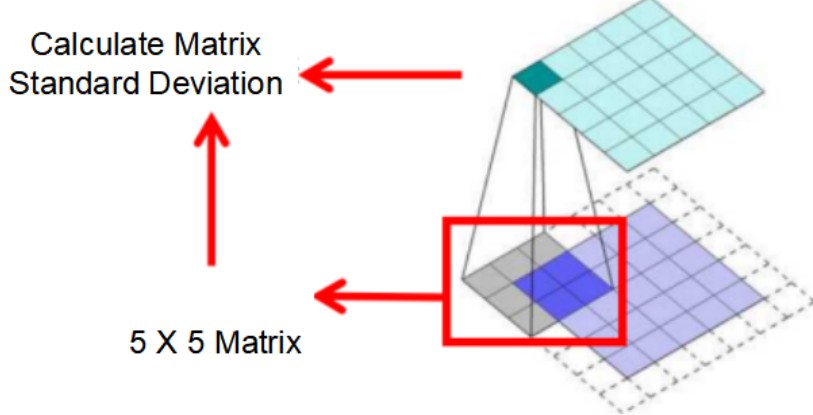

**Figure 3.** Illustration of standard deviation image extraction using convolution processing.

For a visual interpretation, Figure 4a shows a roof image of a building, and Figure 4b shows its standard deviation map. Bright areas include the internal structure of a building and dark areas are regarded as the background.

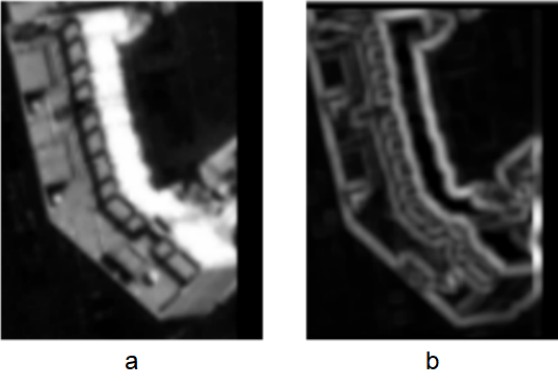

**Figure 4.** Illustration of the image feature map. (**a**) building roof image; (**b**) extracted feature map.

### 2.2. Roof Contour Constrained Stereo Matching Considering Podium Structure

#### 2.2.1. Roof Contour Constrained Stereo Matching

Figure 5 illustrates the epipolar image blocks and their corresponding standard deviation maps. Figure 5a is the backward image block of a building; Figure 5b is the forward image block to be matched. Due to the imaging view difference, these two image blocks have dramatic bright difference. Figure 5c,d respectively shows the standard deviation map of Figure 5a,b, whose bright difference is greatly reduced.

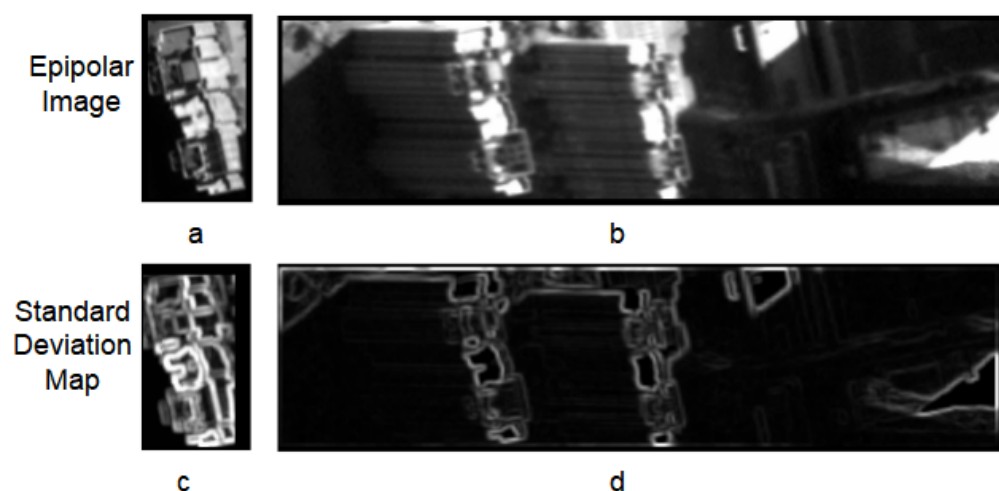

**Figure 5.** Epipolar image blocks and their standard deviation maps. (**a**) backward image block; (**b**) forward image block; (**c,d**) standard deviation maps of image blocks (**a,b**).

Based on the standard deviation maps, we use the gray difference to measure the similarity between the backward image and the forward image. The gray difference similarity is measured as the absolute gray difference between the template and the searched image, as shown in Equation (2), where $g_{S_{i+pj}}$ is the searched pixel value with x parallax p; $w$, $h$ is the template window size; $g_{T_{i+j}}$ is the pixel value of the template window, respectively. The parallax p with the smallest gray difference sum is regarded as the matched result.

$$S(p) = \sum_{i=1}^{w} \sum_{j=1}^{h} \left| g_{S_{i+pj}} - g_{T_{i+j}} \right| \tag{2}$$

In order to exclude the interruption of pixels outside the building roof contours, the gray difference is modified to be a masked gray difference, as shown in Equation (3), in which only the roof pixels are taken into account.

$$S_M(p) = \sum_{i,j \in M} \left| g_{S_{i+pj}} - g_{T_{i+j}} \right| \tag{3}$$

where $M$ is the mask matrix of template block; $(i,j)$ is the pixel in the mask set.

As illustrated in Figure 6, a simple building with only one roof height plane can be matched in one pass, in which the mask is generated with the roof contour indicating the inner part of a building roof, and the histogram of the gray difference image has only one obvious peak.

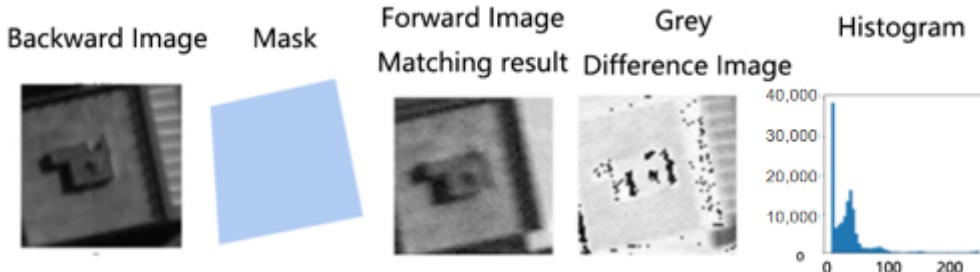

**Figure 6.** A single building matching based on masked gray difference.

### 2.2.2. Segmentation and Rematching of Building with Podium

In the previous section, we suppose that the building roof has only one height. That is correct for most buildings. In the real world, buildings, however, have different shapes and structures, and many buildings have several height planes. Such as, the podium building refers to an auxiliary building that is general at the bottom of the main body of a multi-story, high-rise, or super high-rise building. Its floor area is greater than the standard floor area of the main body. As shown in Figure 7, the pink rectangle and blue rectangle belong to one building. However, they have two different height. For simplicity, we call the building part below the main body as podium building. Apparently, we can not estimate the two heights in one pass matching processing.

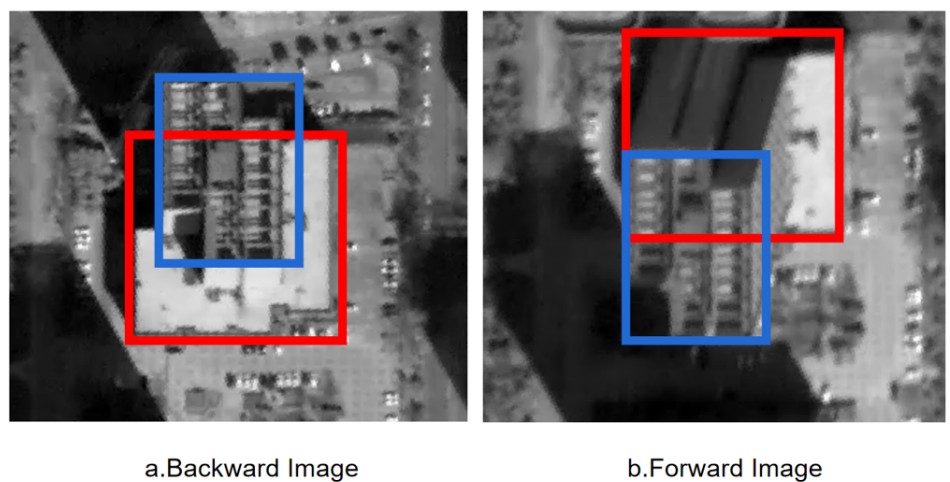

**Figure 7.** Backward and forward view of a podium building. (**a**) podium on backward image; (**b**) podium on forward podium.

Fortunately, the gray difference map of the matched blocks can provide more clues to refine the initial matching result. The gray difference map data represents the average brightness difference of backward and forward parallax. As shown in Figure 6, the gray difference histogram distribution should be in the range of (0–100) values if the matching result is completely correct. However, there are two peaks in the histogram as shown in Figure 8a, one is in the range of (0–100), and the other is in the range of (200–250). This indicates that there are two different heights in the building, and only one height is matched correctly. To solve this problem, we match the different height of a building with podium using the masked gray difference in an iterative manner. As shown in Figure 8a,b, by using two different masks, the bottom height of the podium is estimated in the first pass, and the top height of the podium is estimated in the second pass.

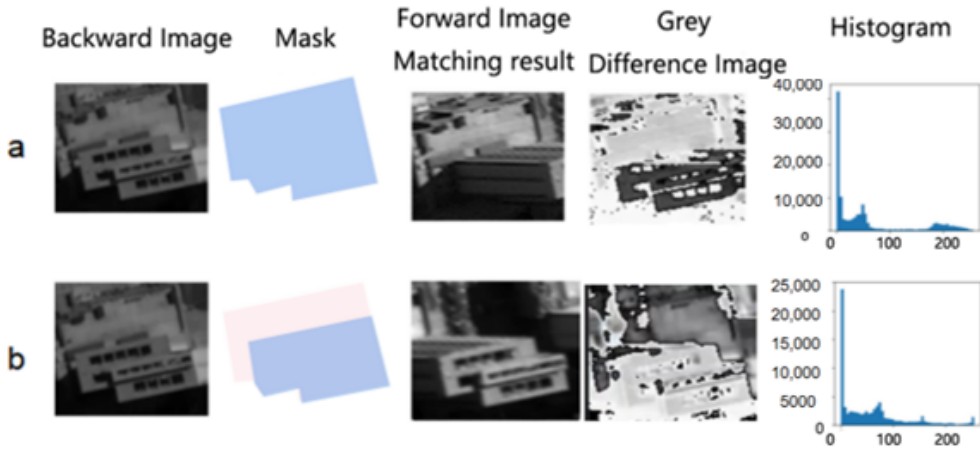

**Figure 8.** Roof matching based on masked gray difference. (**a**) first pass matching for a podium building; (**b**) second pass matching the podium building in (**a**). The blue part of the masks represent the area need to be matched; the pink part of the mask represents the matched area.

Figure 9 shows the principle of mask refinement and contour segmentation of podium building. First, the gray difference image (Figure 9a) is binarized to generate the new mask (Figure 9b). Second, an a connectivity analysis is conducted to erase small regions whose pixel numbers is smaller than a given threshold. The region with maximum number of pixels is selected as the area to be matched (Figure 9c). Third, the morphological filter is carried out on the selected area (Figure 9c) to get a refined mask (Figure 9d). Finally, one or several sub roof contours (Figure 9e) can be generated using the rectangles covering the sub regions of the refined mask, in which the covering rectangles are parallel to the original roof contour.

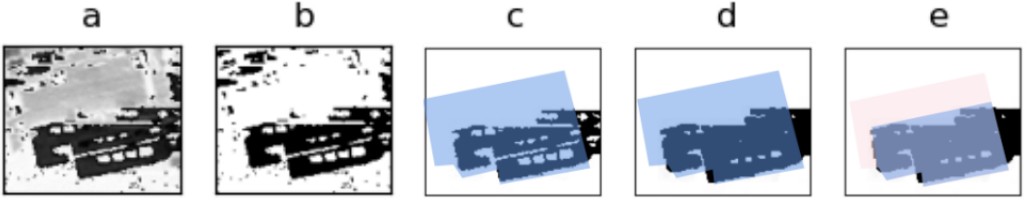

**Figure 9.** Mask refinement and contour segmentation of podium building. (**a**) gray difference map after last pass matching; (**b**) binarization of gray difference map; (**c**) new mask after connectivity analysis; (**d**) refined mask after morphological filtering (**e**) polygon of sub roof contour.

Based on above idea, the process of matching and segmentation of a building with podium is outlined as follows, as illustrated in Figure 10:

(1) Set the mask as the whole roof using the blue contour polygon.
(2) Search the x-parallax with the minimum masked gray difference in the standard deviation map;
(3) Compute the gray difference map between matched image blocks;
(4) Compute the histogram of the gray difference map in the mask set;
(5) If there are more than one peaks in the histogram, the gray difference image is binarized and refined as a new mask for next matching.
(6) Repeat steps 2 to 5 until 90% pixel is matched.

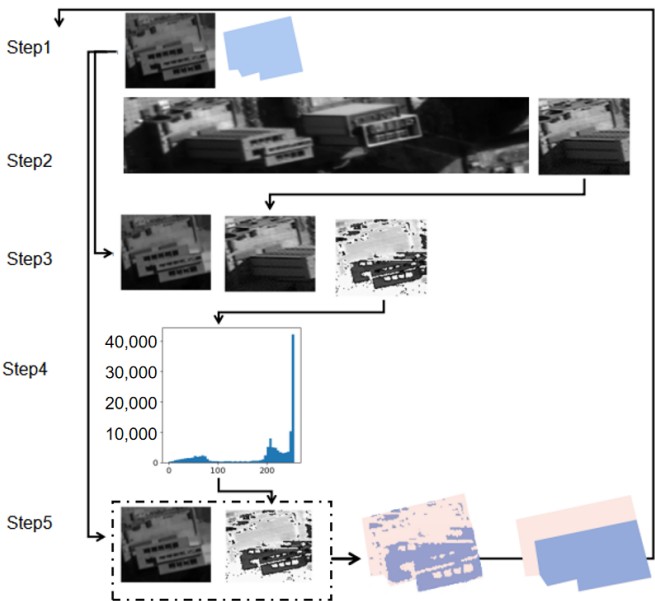

**Figure 10.** Podium Segmentation workflow.

*2.3. Bottom Elevation Estimation and Building Height Calculation*

Three steps are used to estimate the building bottom evaluation and building height. First, combined with the matched x-parallax, the ground point of the building contour center can be computed using the intersection of the two homogeneous imaging rays of the stereo image pair [42]. Then using the RPCs of the backward image, the contour of the building roof on the backward image can be transformed into the object space as the ground footprint, which is on the horizontal plane defined by the elevation of the building center. Second, as shown in Figure 11, the bottom elevation is estimated by using elevation histogram analysis of DSM values, which is created in a 20 m buffer around the building footprint. Then the elevation at the minimum peak of the histogram is chosen as the bottom elevation. Third, the building height is obtained by subtracting the bottom elevation from the roof elevation.

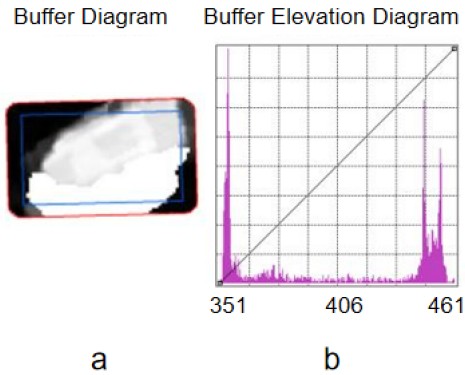

**Figure 11.** Bottom elevation extraction. (**a**) the buffer of DSM; (**b**) the elevation histogram.

**3. Experiment and Results**

*3.1. Data and Study Area*

3.1.1. Study Area

There are two datasets used in this study for performance evaluation, as shown in Figure 12. Dataset one locates in Yingde City, Guangdong Province, between 113.31–113.50°E, 24.25–24.41°N. This study area contains more than 8000 buildings with their footprints in shapefile format, whose height ranges from 10 m to 100 m. By using aerial LiDAR data, we

estimated the roof elevation, bottom elevation, and building height as ground truth. With the GF-7 images, we used ENVI and INPHO to generate DSM, which can be used to estimate building height directly or as the input of our roof contour constrained matching method. Then, our result and DSM based result were compared with the LiDAR based ground truth. In addition, the shadow based result was also compared.

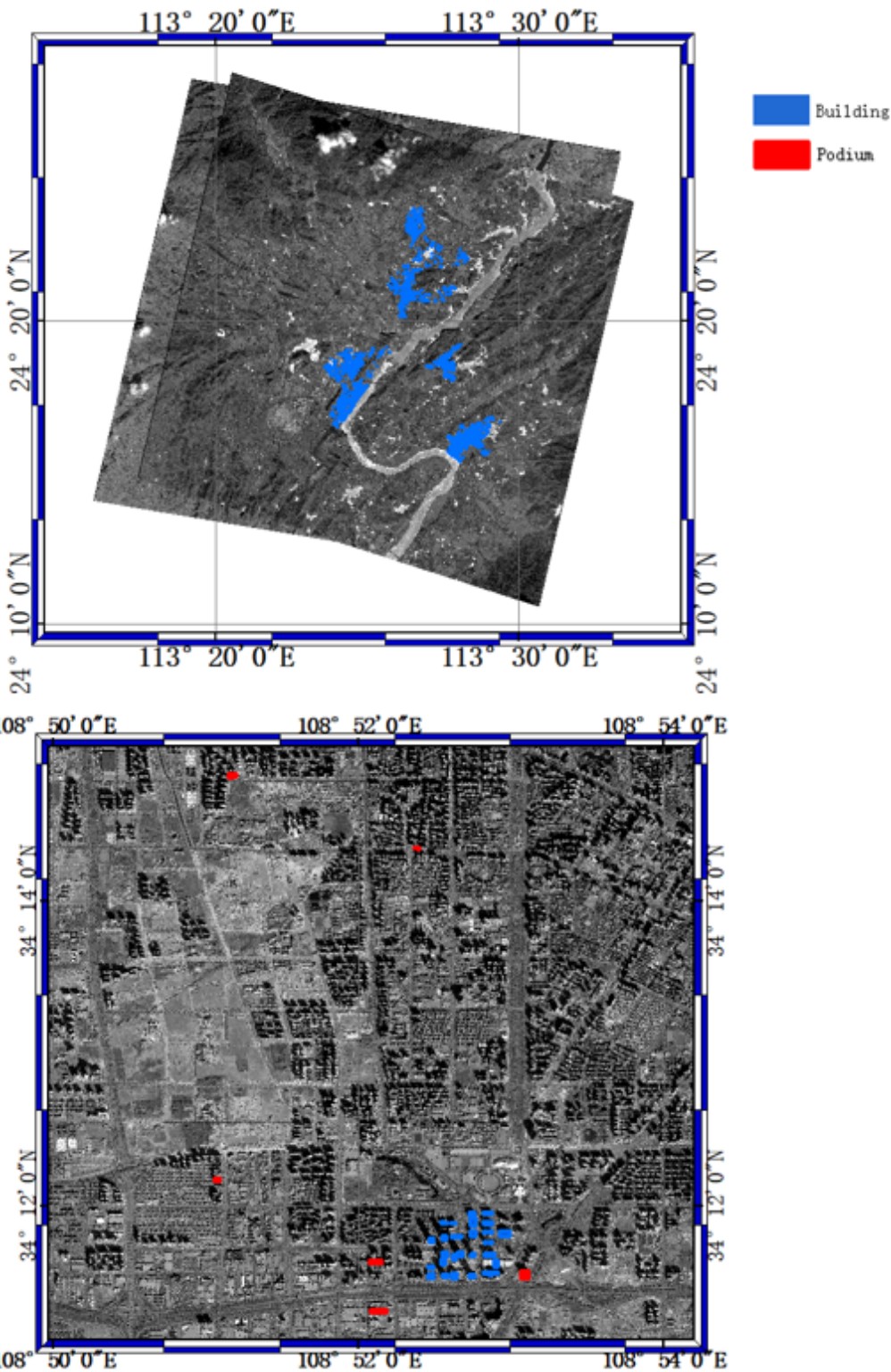

**Figure 12.** GF-7 images and building samples of the study areas.

Dataset two locates in Xi'an high tech Industrial Development Zone, Shanxi Province, between 108.85–108.89°E, 34.18–34.24°N. In this area, we focus on high-rise buildings and podium buildings. In this test site, there are no building footprints and LiDAR data to generate ground truth. Instead, we took the building heights measured from GF-7 stereo images as reference values. We selected 40 buildings, which include 33 single buildings and 7 podium buildings. The height distribution of this dataset ranges from 20 m to 350 m, and the average building height is about 95.55 m, among which the Xi'an Guorui financial center with a height of 348.384 m is the highest building.

### 3.1.2. GF-7 Stereo Images

GF-7 satellite stereo images are used to extract building height. The parameters of used GF-7 satellite stereo images are presented in Table 1. GF-7 satellite is equipped with payloads, including a dual-line array camera and a laser altimeter. The three-line array camera can effectively obtain panchromatic stereo images with a width of 20 km and a resolution better than 0.8 m. In addition, infrared multispectral images with a resolution of 2.6 m can also be obtained. The infrared multi-spectral camera is aligned with the backward panchromatic camera for pansharpening. The backward direction is nearly nadir, designed for balance between less occlusion and bigger stereo intersection angle. Thus, the backward image is better for the extraction of building roof contours.

**Table 1.** Characteristics of the GF-7 satellite stereo images.

| Parameter | Value |
| --- | --- |
| Forward camera inclination | 26° |
| Backward camera inclination | 5° |
| Panchromatic resolution | Backward 0.65 m, Forward 0.8 m |
| Multispectral resolution | Backward 2.6 m |
| Width of Windows | ≥20 km |

### 3.1.3. LiDAR Data

The LiDAR data used in this paper was acquired with Leica CityMapper-2 [43], which is an airborne hybrid sensor of oblique camera and LiDAR. The flight altitude is about 1450 m and the point cloud density in Yingde is 8.1 points/m$^2$. As an example, a small area of LiDAR point cloud in Yingde is shown in Figure 13.

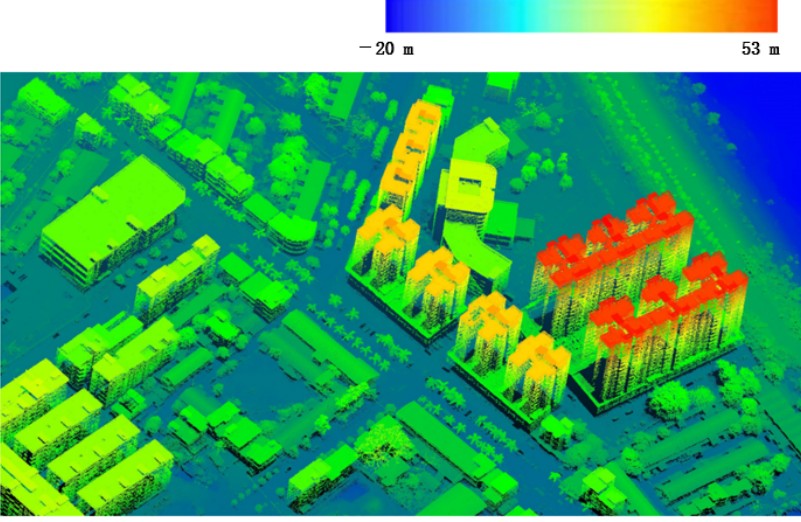

**Figure 13.** A small area of LiDAR point cloud in Yinde.

### 3.1.4. Building Roof Contours and Ground Truth

Our method supposes that building roof contours on the backward image are available for matching against the forward image. For the experiment purpose, building roof contours are generated from footprint data or digitized directly on the backward image. In the practical processing of city scale data, the building contour can be generated by semantic segmentation of backward image.

In the Yingde study area, roof contours are based on the building footprint, which were provided by Hubei Sunrise Photogrammetric Company. Based on the histogram analysis of the point cloud in the buffer of a building footprint, the roof elevation and bottom elevation of a building can be estimated, then the building footprint with roof elevation can be projection onto the backward image as the input roof contour for our workflow. Partial roof contours are shown in Figure 14a.

In the Xi'an study area, we directly digitized 40 buildings' roof contours on the backward image with the ArcMap software, as shown in Figure 14b.

The ground truth of building height was calculated according to the roof elevation and bottom elevation. As mentioned ahead, the roof elevation and bottom elevation in the Yingde study area were estimated with LiDAR data. In the Xi'an study area, the roof elevation and bottom elevation were measured in a self-developed stereo model viewer software.

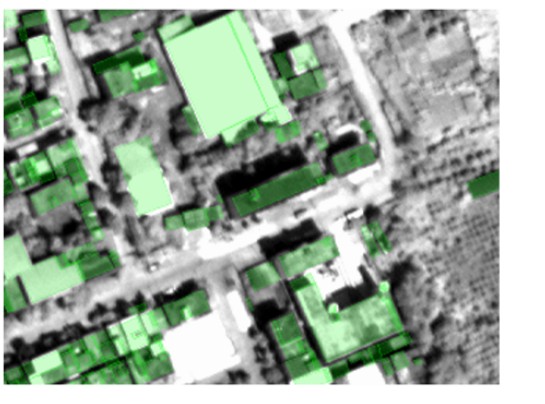 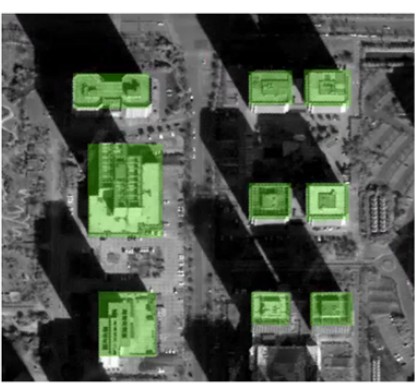

a.Yingde        b.Xi'an

**Figure 14.** Self-annotated building roof contours on the GF-7 backward image. (**a**) the building roof contours in Yingde; (**b**) the countours in Xi'an.

### 3.1.5. DSM Generation

The input DSM in the workflow(shown in Figure 1) is generated by a self-developed dense matching software based on the pyramid semi-global algorithm, which is silimar to SURE [44].

The DSM used in the Section 3.3.2 is generated by commercial software ENVI and INPHO, which are industry leading software.

As shown in Figure 15, five control points were selected from the LiDAR point cloud and the positioning accurarcy of GF-7 stereo image pair was imporved with RPC modification [45] before DSM generation.

For the ENVI, the parameters are set as follows: the minimum overlap is 55; the matching threshold is 15; the edge threshold is 5; the quality threshold is 60; the terrain type is Flat.

For INPHO generating DSM, the parameters are set as follows: the terrain type is Flat; the feature density is Dense; the point cloud density is 3 pixels; the Parallax threshold is 20 pixels. Three block of DSM results generated from ENVI and INPHO are shown in Figure 16.

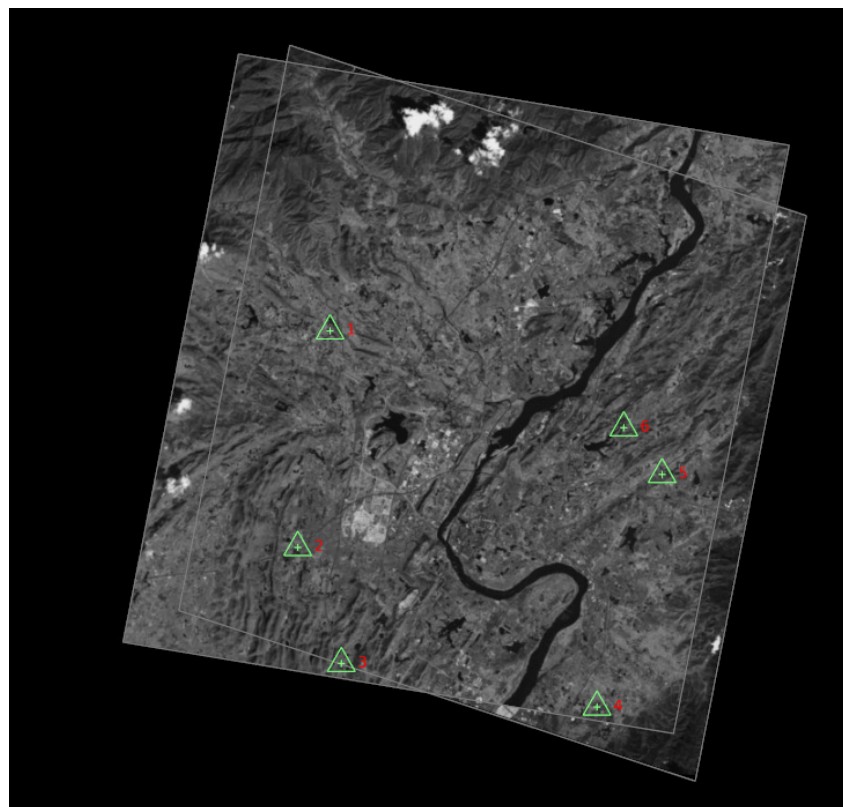

**Figure 15.** Ground control points selected for the Yingde stereo image pair.

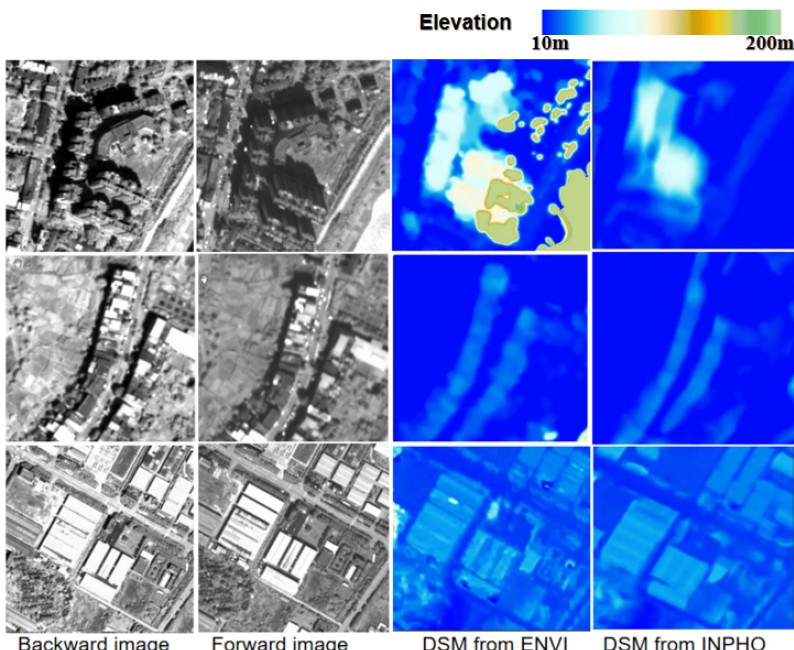

**Figure 16.** Epipolar image blocks and the DSM generated by ENVI and INPHO.

### 3.2. Evaluation Metrics

Height accuracy is evaluated by comparing the estimated building height and the reference building height. Three metrics are used for performance evaluation, including the mean absolute error (MAE), root mean square error (RMSE), and max absolute error (maxAE), which are formulated as Equations (4)–(6):

$$\text{MAE} = \frac{1}{N}\sum_{i=1}^{N}\left|h_i - \hat{h}_i\right| \tag{4}$$

$$\text{RMSE} = \sqrt{\frac{1}{N}\sum_{i=1}^{N}\left(h_i - \hat{h}_i\right)^2} \tag{5}$$

$$max\text{AE} = max\{|h_i - \hat{h}_i|, i = 1, 2, \ldots, N\} \tag{6}$$

where, $h_i$ is the estimated building, $\hat{h}_i$ is the reference building height.

### 3.3. Experimental Results of Yingde Dataset

In Yingde study area, there are a total number of 8653 buildings, whose height is less than 100 m. For accuracy evaluation, the ground truth of building height is generated from aerial LiDAR data. Table 2 presents the accuracy statistic of building height estimation. The results show that the MAE of the roof and bottom elevation estimation are 1.57 m and 1.91 m, respectively, and the accuracy of building height estimation is 2.31 m. The RMSE of building height is 3.01 m. Besides, Table 3 shows the number and accuracy statistic of building height at varying elevation ranges. The results show that the overall building roof elevation estimation is relatively stable, and the bottom elevation estimation accuracy fluctuates wildly since the DSM is easily affected by the occlusion of trees and buildings.

**Table 2.** Accuracy statistic of building height estimation.

| Index | Roof Elevation | Bottom Elevation | Building Height |
|---|---|---|---|
| MAE (m) | 1.57 | 1.91 | 2.31 |
| RMSE (m) | 2.01 | 2.57 | 3.01 |
| maxAE (m) | 12.67 | 4.99 | 9.97 |

**Table 3.** Number and accuracy statistic at varying elevation ranges.

| Height Range | Building Numbers | MAE | | |
|---|---|---|---|---|
| | | Roof Elevation | Bottom Elevation | Building Height |
| 0–10 m | 5774 | 1.56 | 1.66 | 2.05 |
| 10–20 m | 2466 | 1.59 | 2.32 | 2.76 |
| 20–30 m | 305 | 1.48 | 2.73 | 3.07 |
| 30–40 m | 48 | 1.56 | 3.29 | 3.75 |
| 40–50 m | 21 | 1.29 | 4.39 | 4.85 |
| 50–60 m | 17 | 1.12 | 3.20 | 3.65 |
| 60–70 m | 8 | 2.27 | 1.69 | 2.71 |
| 70–80 m | 8 | 1.24 | 1.47 | 1.98 |
| 80–90 m | 6 | 0.76 | 3.67 | 2.90 |
| 90–100 m | 1 | 1.99 | 4.31 | 6.31 |

#### 3.3.1. Comparison with the Shadow-Based Method

The proposed solution is compared with the shadow-based method in this section. Two sub datasets that are extracted from Yingde dateset, are used for evaluation. The first sub dataset is a low-rise building dataset with the height ranging between 0–30 m, and there are 299 buildings in the low-rise building dataset; the second sub dataset is a high-rise building dataset with the height larger than 30 m, and there are 170 buildings in the high-rise building dataset. According to Xie [20], when the ground height is horizontal, the shadow length of a building is directly proportional to the height of the building. Based on this observation, we measure the shadow length of buildings manually to avoid the deficiencies of automatic extraction methods. Because the inclined angles of forward camera and backward camera are different in GF-7 satellite, we estimate the building height on both the forward image and the backward image for extensive comparison.

The statistic results of shadow-based method and the proposed solution are shown in Tables 4 and 5. Table 4 shows the comparison of accuracy statistics of building height. According to the experimental results, we can conclude that: (1) the MAE of the shadow estimation method on the high-rise building dataset is less than that of the low-rise building; (2) In low-rise building dataset, due to the occluded image in the shadow area, 10% buildings in the forward and 36% buildings in the backward participate in the metric calculation. The average accuracy of the proposed solution in terms of MAE, RMSE and maxAE is significantly higher than that of the shadow-based method; (3) In high-rise building dataset, 45% the buildings in forward and 59% the buildings in backward participate in the calculation. Similarly, the accuracy of the proposed solution is significantly higher than that of the shadow-based method for the three metrics.

In addition, Table 5 shows the comparison of number statistic of building height at varying elevation ranges, in which the term occlusion indicates the number of buildings that occluded. we can see that for the low-rise building dataset, 268 (89%) buildings have shadow occlusions on the forward image, and 190 (64%) buildings have shadow occlusions on the backward image; for the high-rise building dataset, 93 (55%) buildings have shadow occlusions on the forward image, and 69 (40%) buildings have shadow occlusions on the backward image. The height of all the occluded buildings cannot be estimated. On the contrary, the proposed method can successfully estimate the height of all buildings.

**Table 4.** The comparison of accuracy statistic of building height.

| Para. | Building < 30 m | | | Building > 30 m | | |
|---|---|---|---|---|---|---|
| | Forward | Backward | Ours | Forward | Backward | Ours |
| MAE (m) | 2.63 | 3.02 | 1.34 | 3.72 | 3.09 | 1.43 |
| RMSE (m) | 3.48 | 14.93 | 1.77 | 4.60 | 6.97 | 1.90 |
| maxAE (m) | 8.11 | 13.50 | 4.75 | 9.59 | 8.59 | 4.63 |

**Table 5.** The comparison of building number statistic at varying height ranges.

| AE | Building < 30 m | | | Building > 30 m | | |
|---|---|---|---|---|---|---|
| | Forward | Backward | Ours | Forward | Backward | Ours |
| <2 (m) | 16 | 53 | 229 | 25 | 38 | 127 |
| 2–4 (m) | 8 | 22 | 65 | 25 | 23 | 30 |
| 4–6 (m) | 3 | 14 | 5 | 7 | 12 | 13 |
| >6 (m) | 4 | 20 | 0 | 20 | 28 | 0 |
| occlusion | 268 | 190 | 0 | 93 | 69 | 0 |

For a visual interpretation, Figure 17 illustrates some shadow occlusion cases: 1 Sheltered by trees; 2 blocked by nearby buildings. As shown in Figure 17a,c, the shadows at the edge of the house are blocked by trees, which belongs to case 1; On the backward image in Figure 17b, the shadow is not obscured due to the small inclination of the camera; however, in Figure 17d, the shadow on the forward image is occluded by the buildings in the backward and forward rows. It can be seen that the shadow-based method has obvious limitations when applied to GF-7 images. In a word, a majority of buildings cannot be extracted by using the shadow-based method due to the serious occlusion that occur in GF-7 satellite images, especially for the forward image.

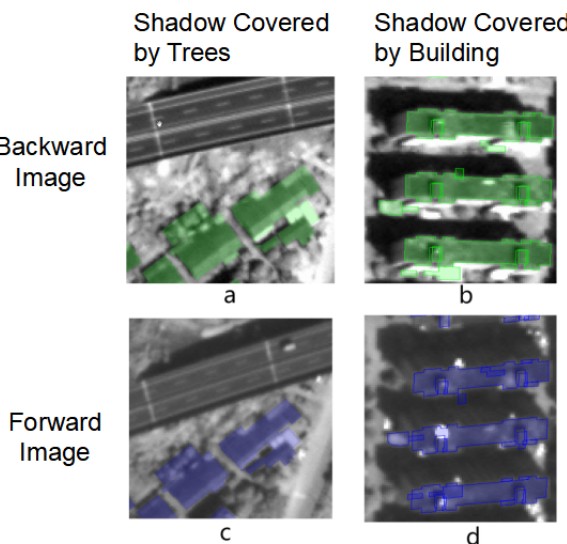

**Figure 17.** The illustration of shadow occlusions. (**a**,**c**) building shadows occluded by trees; (**b**,**d**) building shadows occluded by buildings.

### 3.3.2. Comparison with DSM Based Method

In this section we compare the proposed solution with DSM Based Method. We used ENVI (version 5.6) and INPHO (version 12.1) to generate DSM in Yingde. After generating the DSM, we overlaped the building contour vector onto the DSM for building height extraction. The roof elevation and the bottom elevation are estimated with histogram analysis, which has been discussed in the Section 2.3. The peaks of maximum elevation and minimum elevation are selected as of the building roof and the bottom. Shown in Figure 16, DSM generated by INPHO is relatively smooth, however, it loses some high-rise heights. On the contrary, the DSM generated by ENVI can retain high-rise buildings better, however, it can contain some holes.

Table 6 lists the accuracy statistic of building roof elevation from the DSM based methods, and the distribution of absolute error (AE) of ENVI and INPHO is presented in Figure 18. The results show that ENVI achieves better accuracy than INPHO for building roof height estimation in the two datasets. For both ENVI and INPHO, the estimation accuracy of low-rise buildings is better than that of high-rise buildings. As was expected, our method achieves the highest precision. In a conclusion, INHPO and ENVI have limitations in roof estimation of high-rise buildings when applied to GF-7 images, and the performance of our method is very stable for both high and low buildings. Figure 19 shows the 3D reconstruction models of the proposed solution in this dataset.

**Table 6.** Accuracy statistics of building roof elevation.

| Para. | Building < 30 m | | | Building > 30 m | | |
|---|---|---|---|---|---|---|
| | **ENVI** | **INPHO** | **Ours** | **ENVI** | **INPHO** | **Ours** |
| MAE (m) | 5.12 | 6.53 | 1.34 | 15.6 | 32.65 | 1.43 |
| RMSE (m) | 6.13 | 8.64 | 1.77 | 28.19 | 38.70 | 1.90 |
| maxAE (m) | 26.67 | 35.95 | 4.75 | 82.87 | 89.86 | 4.63 |

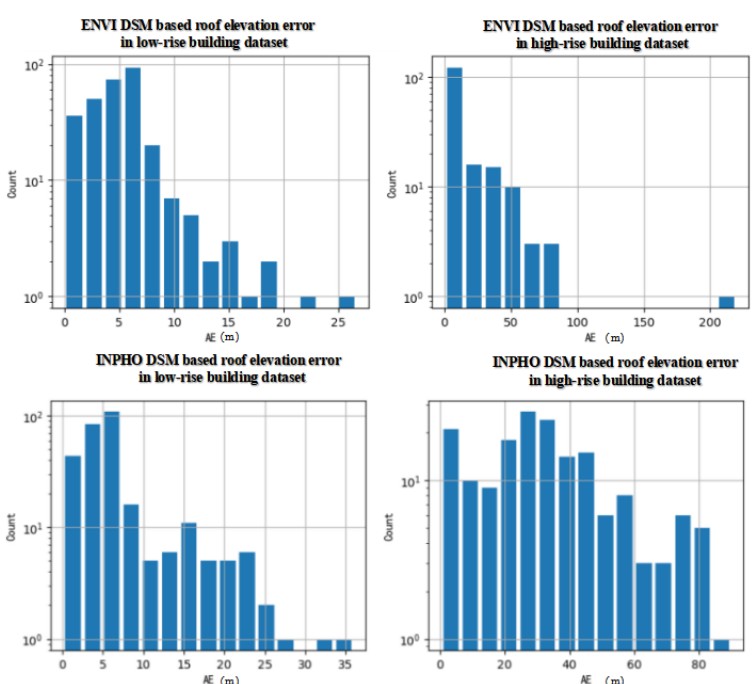

**Figure 18.** Error distribution of DSM based roof elevation.

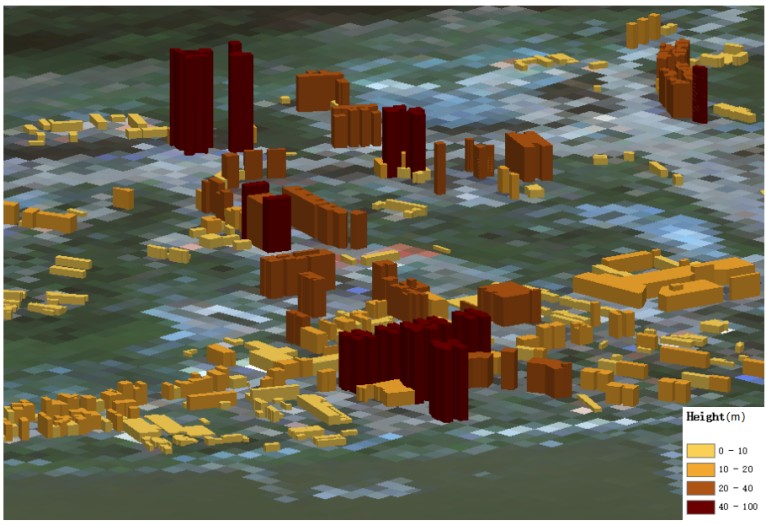

**Figure 19.** The 3D reconstruction models of the proposed solution in Yingde.

### 3.4. Experimental Results of Xi'an Dataset

In the dataset of Xi'an, there are a total number of 40 buildings, in which 7 buildings are with podium. Similar to the procedure presented in Section 3.3.2, the proposed method is compared with DSM based results of ENVI and INPHO. Table 7 shows the accuracy statistic for this dataset. Figure 20 shows the correlation the estimated building heights and their reference values, in which the dots nearer the dashed line have the higher estimation precision. From the results listed in Table 7, we can conclude that our method achieves the best performance, and obviously improves the accuracy of building height estimation; its MAE and RMSE of building height estimation is 1.69 m and 2.34 m, respectively. By further observation from Figure 20, we can see that within the range of 0–100 m, ENVI can achieve comparable accuracy when compared with the proposed method. However, with the increase of building height, its performance dramatically decrease. It can be seen from the

green dots, which are nearly on the dashed line, our method archives consistently high precision within the entire building height range.

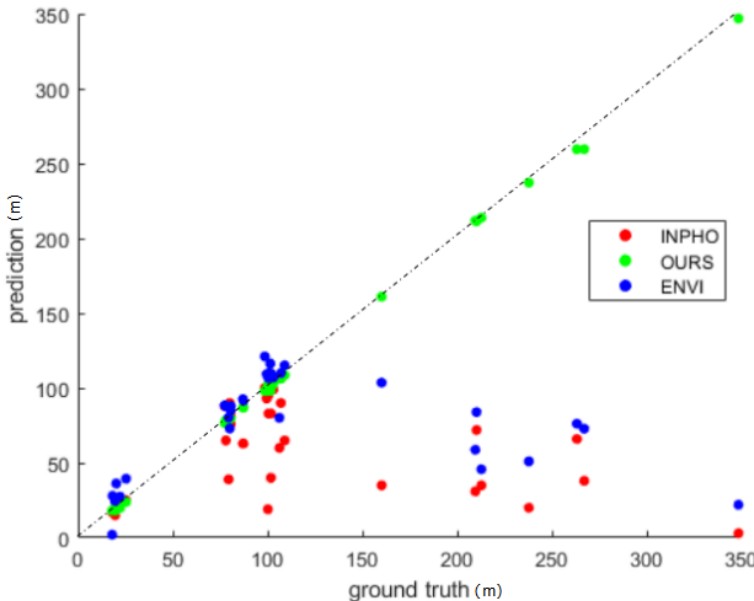

**Figure 20.** The distribution of estimated and ground truth building elevation.

**Table 7.** Accuracy statistics of experimental results in the Xi'an.

| Index | Building Height | | |
|---|---|---|---|
| | ENVI | INPHO | Ours |
| MAE (m) | 41.46 | 55.20 | 1.69 |
| RMSE (m) | 77.86 | 100.64 | 2.34 |
| maxAE (m) | 326.38 | 345.38 | 7.47 |

Building Height Estimation of Buildings with Podium

In the Xi'an dataset, the building heights of the 7 podium buildings are evaluated, in which more than one roof elevations are extracted using our iterative roof matching method. For performance evaluation, the height of each podium building is divided into three parts, and their heights are measured manually from GF-7 stereo image as the reference values.

Table 8 shows the statistical results of building height estimation, which lists the value of the first estimated building height by using the term Whole and height for each part after the podium segmentation by using the terms Part1, Part2 and Part3. Besides, the empty values in Table 8 represent that the part of the second podium is completely covered by other parts, and the ground truth building height cannot be estimated, as shown in Figure 21. The results show that for each part of the building with podium, the proposed method by using podium segmentation can find several elevation planes in a building contour, and can also accurately estimate their height. By the further analysis, we conclude that the main error of height estimation comes from the matching error caused by partial or entire occlusion of building roofs. Figure 22 shows the 3D reconstruction models of the proposed solution in this dataset.

**Table 8.** Accuracy statistics of Podium building results in the study area.

| Index | Whole Ours (m) | Part1 Truth (m) | Part1 Ours (m) | Part2 Truth (m) | Part2 Ours (m) | Part3 Truth (m) | Part3 Ours (m) |
|---|---|---|---|---|---|---|---|
| 1 | 106.19 | 108 | 106.19 | 16 | 18.86 | | |
| 2 | 74.71 | 106 | 105.18 | - | - | | |
| 3 | 60.49 | 75 | 70.65 | 75 | 70.65 | 10 | 9.72 |
| 4 | 17.85 | 52 | 49.33 | 25 | 18.86 | | |
| 5 | 30.04 | 55 | 50.35 | 16 | 13.79 | | |
| 6 | 4.65 | 87 | 86.90 | 12 | 7.70 | | |
| 7 | 4.65 | 93 | 90.65 | 2 | 4.65 | | |

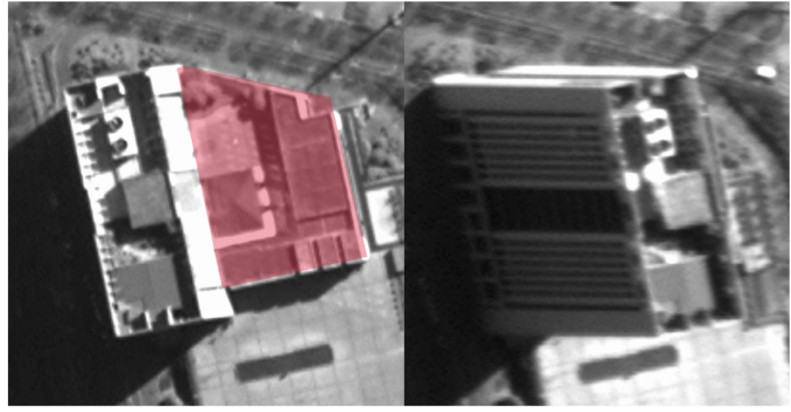

Backward image      Forward image

**Figure 21.** The illustration of the second Podium structure. The red box indicates the coverd part on backward image.

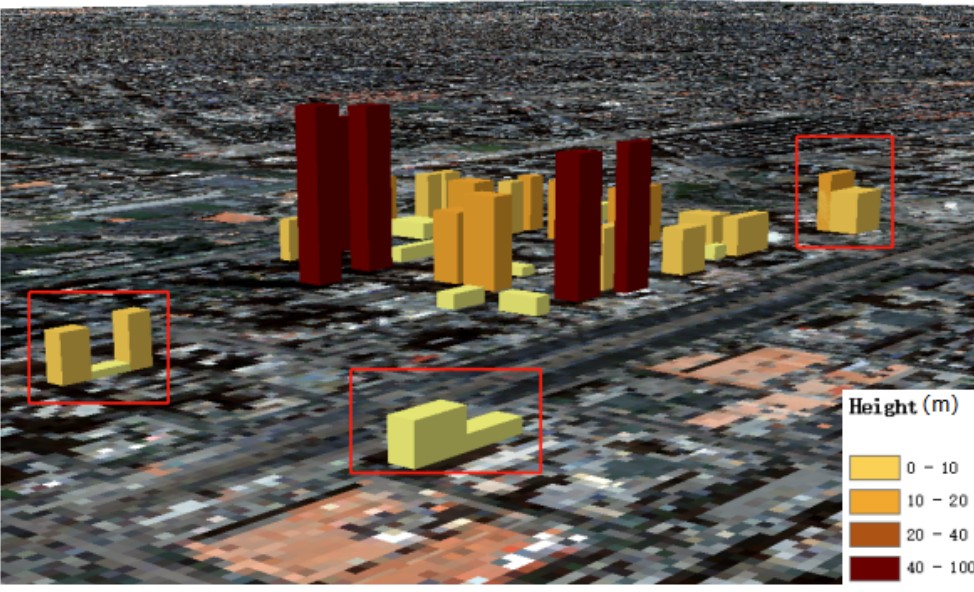

Height (m)

- 0 – 10
- 10 – 20
- 20 – 40
- 40 – 100

**Figure 22.** The 3D reconstruction models of the proposed solution in Xi'an. The red box indicates the podium buildings.

## 4. Conclusions

This study proposes a method for extracting building height information from GF-7 satellite stereo images. First, an object-oriented roof matching algorithm is proposed based on building contour to extract accurate building roof elevation from GF-7 stereo image,

and DSM generated by business software is then used to obtain building bottom elevation. Second, to cope with buildings with multiple level height plane, a mask gray difference metric is proposed to search multiple elevation planes and segment the building contour. Finally, by using ground truth data from LiDAR point clouds or manually measured building height, the proposed solution is extensively evaluated and compared with shadow-based method and DSM based method. The experimental results demonstrate that the proposed solution can achieve the best performance and could be an useful solution for accurate and automatic building height information extraction from GF-7 satellite stereo images.

Compared with the DSM and shadow-based method, our algorithm skillfully solves the problem of height estimation of high-rise buildings. Compared with deep learning, this algorithm does not need training data set. For the first time, we focused on the problem of multiple elevation planes within the height of a building.

**Author Contributions:** Conceptualization, C.Z., W.J. and Y.C.; Methodology, C.Z., W.J. and Y.C.; Software, C.Z. and W.J.; Visualization, C.Z.; Validation, C.Z. and Z.Z.; Formal analysis, C.Z.; Investigation, C.Z.; Resources, W.J.; Data curation, C.Z. and Z.Z.; Writing—original draft preparation, C.Z.; Writing—review and editing, C.Z., W.J. and S.J.; Supervision, W.J.; Project administration, C.Z. and J.W; Funding acquisition, W.J. All authors have read and agreed to the published version of the manuscript.

**Funding:** This research was funded by Fund Project: High-Resolution Remote Sensing Application Demonstration System for Urban Fine Management. Grant number is 06-Y30F04-9001-20/22.

**Data Availability Statement:** Not applicable.

**Conflicts of Interest:** The authors declare no conflict of interest.

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
