# Peer review of "Building Height Extraction from GF-7 Satellite Images Based on Roof Contour Constrained Stereo Matching"

_remotesensing, doi:10.3390/rs14071566_

Round 1

Reviewer 1 Report

The manuscript describes the algorithms and methods for automatic extractions of podiums building based on the GF-7 satellite images and archival DSM. The presented investigation is very interesting for urbanists and civil specialists. Nevertheless, I am afraid that the manuscript needs significant changes.

In my opinion, this paper has a severe structural problem. Despite that, the Introduction introduces the reader to the issue of the performed investigation, the description of the results and comparison to another method should be extended and improved. In general, the main information about the ground-truth data is missed. Please provide more detail about the quality of the LIDAR data used for DSM generation and the parameters used in Inpho and ENVI softwares for data processing. Without it, it is hard to assess the proposed method for building hights determination.

The quality of the figures should be improved and the placement in the text.

Reviewer 2 Report

Using VHR images to estimate the building height is a very important and challenging task. This paper proposed a practical method to calculate the height value of the building from GF-7 image constrained by DSM and roof contour. The method is verified by large number of building instance. The results is very promising. However, I think there are some limits in the paper.

(1) The source of the historical DSM used in this paper are better to be given in more details. If there is some elevation information in the contour, I think it is better to described it.

(2) Line 19, “greater than 2.31 m” is right?

(3) Line 187, why the intersection of the point is the Zmin? If some buildings on the DSM source demolish or shorten?  Or there are some error in the DSM.

(4) In eq(2), what’s value of the w,h in the paper?

(5) Line 284, I think the first “bottom” has to be changed to “roof”.

(6) In figure 2, the roof contours is correctly overlapped on the building outline in the images, but the images are not ortho-images, I think the details of self-annotation processing is important for possible readers.

(7) In my mind, I think the similarity measure based on absolute gray difference is not very stable as the match window between the backward and forward images may be very different due to the large intersection angle and height of the building. The epipolar resampling can’t deal with the geometry distortion between the matching windows. Thus I think, the detail on how to obtain the correct match for each roof has to be given. Just match the center of the roof? To select the most robust result? Or Match the whole roof, but how to deal with the distortion, as the contour only can constrain the template window on one image?

(8) Figure 9, there is no red color, a mistake?

(9) What’s the meaning of different color in the Figure 15.

(10) Some characters in some figures are not very clearly may confuse potential readers.

(11) In the abstraction, I think the results are better to given in the same criteria for both data sets.

Round 2

Reviewer 1 Report

The authors have taken all my comments on board. Nevertheless, I would ask you to explain why Authors decided to"directly digitized 40 buildings' roof contours on the backward image with the ArcMap software"(lines 295 - 296). Does this imply that distortions caused by the central projection have not been taken into account?
